# ViDA: Homeostatic Visual Domain Adapter for Continual Test Time Adaptation

## Abstract

Since real-world machine systems are running in non-stationary and continually changing environments, Continual Test-Time Adaptation (CTTA) task is proposed to adapt the pre-trained model to continually changing target domains. Recently, existing methods mainly focus on model-based adaptation, which aims to leverage a self-training manner to extract the target domain knowledge. However, pseudo labels can be noisy and the updated model parameters are uncertain under dynamic data distributions, leading to error accumulation and catastrophic forgetting in the continual adaptation process. To tackle these challenges and maintain the model plasticity, we tactfully design a Visual Domain Adapter (ViDA) for CTTA, explicitly handling both domain-specific and domain-agnostic knowledge. Specifically, we first comprehensively explore the different domain representations of the adapters with trainable high and low-rank embedding space. Then we inject ViDAs into the pre-trained model, which leverages high-rank and low-rank prototypes to adapt the current domain distribution and maintain the continual domain-shared knowledge, respectively. To adapt to the various distribution shifts of each sample in target domains, we further propose a Homeostatic Knowledge Allotment (HKA) strategy, which adaptively merges knowledge from each ViDA with different rank prototypes. Extensive experiments conducted on four widely-used benchmarks demonstrate that our proposed method achieves state-of-the-art performance in both classification and segmentation CTTA tasks. In addition, our method can be regarded as a novel transfer paradigm and showcases promising results in zero-shot adaptation of foundation models to continual downstream tasks and distributions.

## 1 Introduction

Deep Neural Networks (DNN) have achieved remarkable performance in various computer vision tasks, such as classification [22, 14], object detection [48, 63], and segmentation [9, 58], when the test data distribution is similar to the training data. However, real-world machine perception systems (i.e., autonomous driving [1, 28]) operate in non-stationary and constantly changing environments, which contain heterogeneous and dynamic domain distribution shifts. Applying a pre-trained model in these real-world tasks [50] can lead to significant degradation in perception ability on target domains, especially when the target distribution changes unexpectedly over time. Therefore, developing continual domain adaptation (DA) methods that can enhance the generalization capability of DNNs and improve the reliability of machine perception systems in dynamic environments.

A classical source-free DA task, Test-Time Adaptation [39] (TTA), eases the distribution shift between a source domain and a fixed target domain. This is typically achieved through the utilization of self-training mechanisms [42, 55]. However, when adapting to continually changing target domains, pseudo labels are noisy and the updated model parameters become uncertain, leading to error

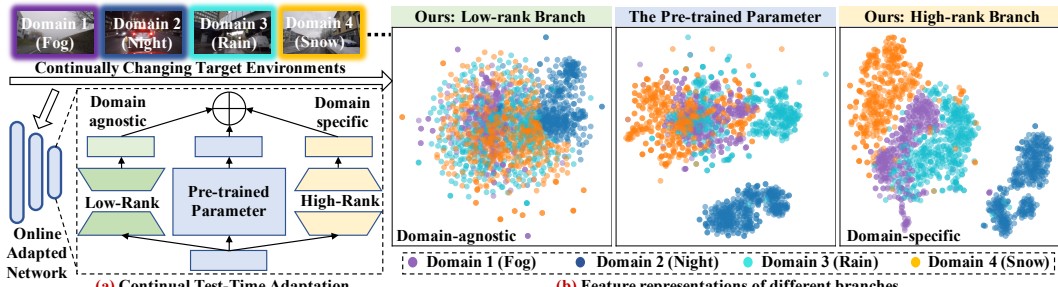

Figure 1: **The problem and motivation of our method.** (a) Our goal is to effectively adapt the source pre-trained model to continually changing target domains. We propose Visual Domain Adapters with different domain representations to tackle the error accumulation and catastrophic forgetting challenges during the continual adaptation process. We leverage ViDAs with high-rank and low-rank prototypes to adapt current domain distribution and maintain the continual domain-agnostic knowledge, respectively. (b) we conduct a t-SNE [53] analysis for the different adapter distributions across four target domains (ACDC). The low-rank branch exhibits a consistent distribution across the target domains, suggesting that it can effectively disregard the impact of dynamic distribution shifts. The high-rank branch demonstrates noticeable distribution discrepancies between the various target domains, suggesting that it primarily focuses on extracting domain-specific knowledge.

accumulation and catastrophic forgetting. To tackle this problem, Continual Test-Time Adaptation (CTTA) has been proposed [57], which addresses a sequence of different distribution shifts over time rather than a single shift as in TTA. Furthermore, CTTA also encompasses the efficient zero-shot adaptation of foundation models to continual downstream tasks or distributions [2, 29].

Existing CTTA works [57, 7, 16, 59] have primarily employed model-based and prompt-based approaches to extract target domain-specific and domain-invariant knowledge simultaneously. However, for model-based methods [57, 7], the noisy pseudo labels are still unreliable and play a limited role in avoiding error accumulation, particularly in scenarios with significant distribution gaps. Meanwhile, prompt-based methods [16, 59] face difficulties in leveraging soft prompts with limited trainable parameters to learn long-term domain-shared knowledge and prevent catastrophic forgetting.

To tackle these limitations and maintain the model plasticity, we tactfully design a homeostatic Visual Domain Adapter (ViDA), shown in Fig. 1 (a), which explicitly manages domain-specific and domain-agnostic knowledge in the continual adaptation process. Specifically, we first carefully explore the different domain representations of ViDAs with trainable high and low-rank embedding space. Our observations reveal that ViDA with a low-rank prototype focuses on domain-agnostic feature representation in different domains. As shown in Fig. 1 (b), the prototype distribution of the adapter neglects the influence of dynamic distribution shifts. Conversely, ViDA with a high-rank prototype concentrates more on extracting domain-specific knowledge, as evidenced by the prototype distribution in different target domains showing an obvious discrepancy. We provide a detailed explanation of the motivations in Section 3.1.

This observation motivates us to inject ViDAs into the pre-trained model, which leverages high and low-dimension prototype to adapt current domain distribution and maintain the continual domain-shared knowledge, respectively. According to the various distribution shift of each sample, we further propose a Homeostatic Knowledge Allotment (HKA) strategy to dynamically fuse the knowledge from each ViDA with different dimension prototypes. In Fig. 1 (b), HKA adaptively regularizes the balance of different feature representations, including original model, domain-specific, and domain-agnostic features. During inference, the different domain-represented ViDAs can be projected into the pre-trained model by re-parameterization [13], which ensures no extra parameter increase and maintain the model plasticity. In addition, through the proposed homeostatic ViDAs, we empower the model with domain generalization ability, which achieves a significant improvement (+7.6%) on the five unseen target domains of ImageNet-C. In summary, our contributions are as follows:

- We carefully study the different domain representations of the adapters with high and low-rank prototypes. And we tactfully design a Visual Domain Adapter (ViDA) for CTTA, explicitly managing domain-specific and domain-shared knowledge to tackle the error accumulation and catastrophic forgetting problem, respectively.

- According to the various distribution shift of each sample in the target domains, we further propose a Homeostatic Knowledge Allotment (HKA) strategy to dynamically fuse the knowledge from each ViDA with different rank prototypes.

- Our proposed approach outperforms most state-of-the-art methods according to the experiments on four benchmark datasets, covering classification and segmentation tasks.

- Our CTTA method provides a novel transfer paradigm and achieves a promising result in zero-shot adapting of foundation models to continual downstream distributions. Meanwhile, we empower the source model with domain generalization ability through the proposed homeostatic ViDAs, achieving a significant improvement on the unseen target domains.

## 2 Related work

### 2.1 Continual Test-Time Adaptation

**Test-time adaptation (TTA)**, also referred to as source-free domain adaptation [6, 34, 40, 60], aims to adapt a source model to an unknown target domain distribution without relying on any source domain data. Recent research has explored self-training and entropy regularization techniques to fine-tune the source model [35, 56, 40, 8]. Tent [56] updates the training parameters in batch normalization layers by minimizing entropy. Recently, there has been a surge of interest in performing Transformer-based TTA works [57, 20, 20]. **Continual Test-Time Adaptation (CTTA)** refers to a scenario where the target domain is not static, presenting additional challenges for traditional TTA methods. The first approach to address this challenging task is introduced in [57], which combines bi-average pseudo labels and stochastic weight reset. While [57, 7] tackles the problem in both classification and segmentation tasks at the model level, [16] introduces the use of visual domain prompts to address the issue at the input level specifically for the classification task. In this paper, we simultaneously focus on both classification tasks and dense prediction tasks.

### 2.2 Parameter-Efficient Fine-Tuning

Recently, Parameter-Efficient Fine-Tuning (PEFT) has gained significant traction within the field of natural language processing (NLP) [30, 26, 25, 61, 37, 27, 19, 23, 54, 45]. Adapter-based models, a form of PEFT, have gained popularity in NLP. They employ bottleneck architecture adapter modules inserted between layers in pre-trained models. During fine-tuning, only these modules are updated. Adapter-based models demonstrate dominant performance over other methods in certain tasks, sometimes surpassing standard fine-tuning [12]. Inspired by NLP, adapters in visual tasks have also received widespread attention. In the initial phases of adapter development, residual adapter modules [46, 47] are proposed to aid in the effective adaptation of convolutional neural networks across multiple downstream tasks. AdaptFormer [10] enhances the ViT [14] model by replacing the original multi-layer perceptron (MLP) block with AdaptMLP. AdaptMLP introduces a trainable down-to-up bottleneck module in a parallel manner, effectively mitigating catastrophic interference between tasks. VL-Adapter [51] improves the efficiency and performance of adapters by sharing low-dimensional layers weights to attain knowledge across tasks. Existing methods, as mentioned, have not addressed the challenges of long-term preservation of domain-agnostic knowledge and timely exploration of domain-specific knowledge amidst continuous unknown domain variations. Consequently, there is an urgent demand for an adapter with different domain representations that can simultaneously tackle the challenges of error accumulation and catastrophic forgetting.

## 3 Method

In Continual Test-Time Adaptation (CTTA), we pre-train the model $q_\theta(y|x)$ on the source domain $D_S = (Y_S, X_S)$ and adapt it on multiple target domains $D_{T_i} = \{(X_{T_i})\}_{i=1}^n$, where $n$ represents the scale of the continual target datasets. The entire process can not access any source domain data and can only access target domain data once. The distributions of the target domains (i.e., $D_{T_1}, D_{T_2}, ..., D_{T_n}$) are constantly changing over time. Our goal is to adapt the pre-trained model to target domains and maintain the perception ability of the model on the seen domain distribution.

Our approach proposes a novel Visual Domain Adapter (ViDA) that contains both high and low-dimensional prototypes. This design allows us to explicitly manage domain-specific and domain-

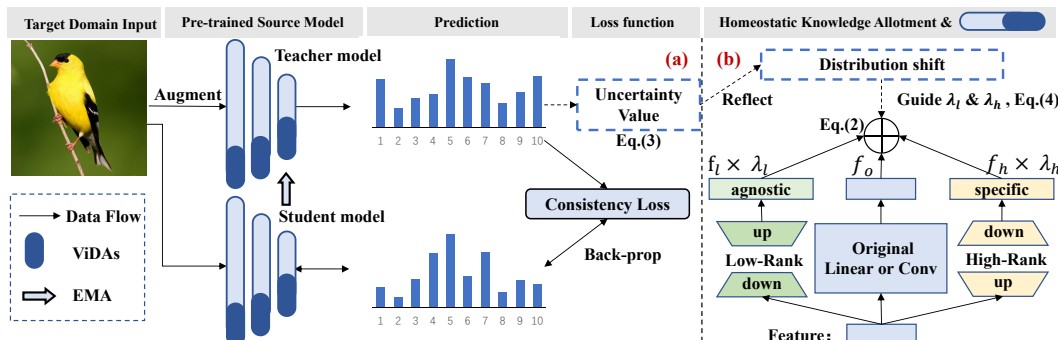

Figure 2: **The framework of Visual Domain Adapter (ViDA).** (a) We inject different domain-represented ViDAs into either linear or Conv layers of the pre-trained source model. To update the ViDAs, we construct a teacher-student framework and use a consistency loss (Eq. 5) as the optimization objective. The student model processes the original image, while the teacher model processes an augmented version of the same image. In addition to generating predictions, the teacher model calculates an uncertainty value (Eq. 3), reflecting the distribution shift of each sample in the target domain. (b) We illustrate the details of the Homeostatic Knowledge Allotment (HKA) strategy, which aims to dynamically fuse the knowledge from each ViDA with different rank prototypes.

agnostic knowledge, addressing the challenges of error accumulation and catastrophic forgetting in CTTA. To effectively adapt to the diverse distribution shifts, a Homeostatic Knowledge Allotment (HKA) strategy is introduced to dynamically fuse the knowledge from different ViDA with different domain representations. The overall framework is shown in Fig. 2.

## 3.1 Motivation

The Continual Test-Time Adaptation (CTTA) faces significant challenges, primarily due to error accumulation and catastrophic forgetting [57, 16]. Meanwhile, adapters with different dimension prototypes demonstrate remarkable effectiveness in addressing these challenges. This encourages us to take a step further and investigate the principles underlying the use of domain adapters in CTTA.

**Adapter with low rank prototype.** Our hypothesis regarding the effectiveness of adapters in mitigating catastrophic forgetting is that their low-rank prototype representation plays a crucial role. To explore this further, we conduct a t-SNE study [53] on the third transformer block to analyze the feature distributions across four target domains (ACDC). The results are depicted in Fig. 1 (b). Our analysis reveals that the low-rank adapter exhibits a relatively consistent distribution across the different target domains, suggesting that its low-rank prototype can effectively disregard the impact of dynamic distribution shifts and prioritize the extraction of domain-invariant knowledge.

We adopt the domain distance definition proposed by Ben-David [4, 3] and build upon previous domain transfer research [18] by employing the $\mathcal{H}$-$divergence$ metric to further evaluate the domain representations of adapters across different target domains. $\mathcal{H}$-$divergence$ between $D_S$ and $D_{T_i}$ can be calculated as $d_{\mathcal{H}}(D_S, D_{T_i}) = 2 \sup_{\mathcal{D} \sim \mathcal{H}} |\Pr_{x \sim D_S}[\mathcal{D}(x) = 1] - \Pr_{x \sim D_{T_i}}[\mathcal{D}(x) = 1]|$, where $\mathcal{H}$ denotes hypothetical space and $\mathcal{D}$ denotes discriminator. Similar to [18], calculating the $\mathcal{H}$-divergence directly is challenging. We adopt the $Jensen$-$Shannon$ $(JS)$ $divergence$ between two adjacent domains as an approximation. To investigate the effectiveness of adapters in adapting to continual target domains, we compare the $JS$ values obtained by using the source model alone, injecting low-rank adapter, and combining low-high adapters, as illustrated in Fig. 3 (a). Our results indicate that the feature representation generated by the low-rank adapter exhibits lower divergence compared to those of the original source model and closely resembles the values of low-high combination.

To provide clearer evidence for our assumption, we have developed an evaluation approach that directly reflects the extent of domain catastrophic forgetting. Shown in Table 1, after one round of CTTA on all target domains (ImageNet-C), we utilize the model and adapter from the last target domain to directly test on previously seen target domains. As expected, the performance degradation is observed in only 2 out of 15 corruption types, and there is an overall improvement of 1.0% in the average classification error. These findings further support our assumptions and indicate that low-rank adapters are more effective in preserving continual domain-shared knowledge.

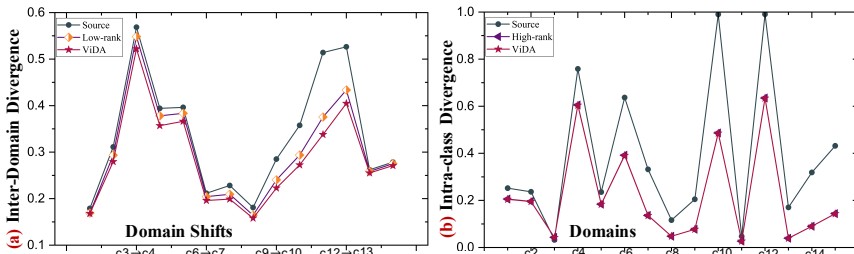

Figure 3: c1 to c15 represent the 15 corruption domains in CIFAR10C listed in sequential order. (a) Low-rank adapter based model effectively mitigates inter-domain divergence than the source model across all 14 domain shifts. (b) High-rank adapter based model significantly enhances the intra-class feature aggregation, yielding results that closely approximate those achieved by our ViDA method.

**Adapter with high rank prototype.** Regarding the domain representation of the adapter with a high-rank prototype, we propose that it is better suited to address error accumulation in the continual adaptation process. We verify this by visualizing the prototype distributions between different domains, as shown in Fig. 1 (b), and observe that there is a clear discrepancy between domains. And the distribution achieves a better aggregation in a single domain. This suggests that high-rank adapters primarily focus on extracting domain-specific knowledge in continual target domains. Inspired by intra-cluster dissimilarity proposed by $k$-means [41], we use normalized intra-class divergence to further verify the domain representations of high-rank adapters in CIFAR10C. As illustrated in Fig. 3 (b), the high-rank adapter is found to drive down divergence within almost all domains, indicating that it can better adapt to current domain distribution and extract domain-specific knowledge in continual target domains. To straightforwardly measure it, we quantitatively evaluate its performance. As shown in Table 6 $Ex_2$, the classification error rate exhibits a sustained reduction (-4.6%) in the dynamic target domains with the use of a high-rank adapter. This finding supports our hypothesis that high-rank adapters can extract more reliable domain-specific knowledge.

## 3.2 Visual Domain Adapter

The above observation motivates us to introduce high-rank and low-rank Visual Domain Adapters (ViDAs) into the source pre-trained model, aiming to simultaneously adapt current domain distribution and maintain the continual domain-shared knowledge in CTTA.

**The architecture.** The design principle of injecting ViDAs into the pre-trained model is simple yet effective, which is illustrated in Figure .2 (b). As we can see there are three sub-branches, the linear (or Conv) layer in the middle branch is identical to the original network, while the right branch and left branch are bottleneck structures and separately indicate the high-rank ViDA and low-rank ViDA. Specifically, the right branch (high-rank) contains an up-projection layer with parameters $W_{up}^h \in R^{d \times d_h}$, a down-projection layer with parameters $W_{down}^h \in R^{d_h \times d}$, where $d_h$ (i.e., $d_h = 128$) is the middle dimension of high-rank prototype and satisfies $d_h \geq d$. There is not any non-linear layer in the ViDA. And we utilize the linear layer as the projection layer when the original model is transformer architecture and adopt $1 \times 1$ Conv as the projection layer when the original model is a convolution network. In contrast, the left branch (low-rank) first injects a down-projection layer with parameters $W_{down}^l \in R^{d \times d_l}$, then place an up-projection layer with parameters $W_{up}^l \in R^{d_l \times d}$, where $d_l$ (i.e., $d_l = 1$) stand for the middle dimension of the low-rank prototype ($d_l \ll d$). For a input feature $f$, the produced features of high-rank ViDA ($f_h$) and low-rank ViDA ($f_l$) are formulated as:

$$f_h = W_{down}^h \cdot (W_{up}^h \cdot f); \quad f_l = W_{up}^l \cdot (W_{down}^l \cdot f) \quad (1)$$

The two-branch bottleneck is connected to the output feature of the original network ($f_o$) through the residual connection via scale factors ($\lambda_h$ and $\lambda_l$). The fusion knowledge ($f_f$) can be described as:

$$f_f = f_o + \lambda_h \times f_h + \lambda_l \times f_l \quad (2)$$

The domain knowledge scale factors ($\lambda_h$ and $\lambda_l$) are adaptively obtained through the homeostatic knowledge allotment strategy, which is shown in Section 3.3.

**Continual adapting.** During the continual adaptation process, we freeze the parameters of the original model (middle branch) and update the high-rank ViDA and low-rank ViDA on the dynamic target domains with unsupervised loss. During inference, the different domain-represented ViDAs (linear relation) can be projected into the pre-trained model by re-parameterization [13], which ensures no extra parameter increase and maintain the plasticity of the original model.

### 3.3 Homeostatic Knowledge Allotment

**Method motivation.** In CTTA, the target domain data can only be accessed once and show different distribution shifts, which makes the efficiency of domain transfer crucial. Moreover, to tackle error accumulation and catastrophic forgetting effectively, it becomes necessary to extract different domain knowledge and handle them separately. This requires regularization of the knowledge fusion weight to ensure efficient capture of relevant domain-specific knowledge without sacrificing the retention of long-term domain-shared knowledge. **HKA design.** As depicted in Figure .2 (b), we draw inspiration from [44, 49, 17] and introduce an uncertainty value to quantify the degree of distribution shift for each sample. While the confidence score is a common measure to assess prediction reliability, it tends to fluctuate irregularly and becomes unreliable in scenarios characterized by distribution shifts. To address this limitation, we employ the MC Dropout technique [15] on linear layers, enabling multiple forward propagations to obtain $m$ sets of probabilities for each sample. Subsequently, we calculate the uncertainty value $\mathcal{U}(x)$ for a given input $x$, which are formulated as:

$$\mathcal{U}(x) = \left( \frac{1}{m} \sum_{i=1}^{m} \|p_i(y|x) - \mu\|^2 \right)^{\frac{1}{2}} \tag{3}$$

Where $p_i(y|x)$ is the predicted probability of the input $x$ in the $i^{th}$ forward propagation and $\mu$ is the average value of $m$ times prediction. To dynamically adjust the scale factors ($\lambda_h$ and $\lambda_l$) based on the uncertainty score, the formulation is as follows:

$$\begin{cases} \lambda_h = 1 + \mathcal{U}(x) & \lambda_l = 1 - \mathcal{U}(x), & \mathcal{U}(x) \geq \Theta \\ \lambda_h = 1 - \mathcal{U}(x) & \lambda_l = 1 + \mathcal{U}(x), & \mathcal{U}(x) < \Theta \end{cases} \tag{4}$$

The threshold value of uncertainty is denoted as $\Theta$, where $\Theta = 0.2$. To realize the homeostasis of different domain knowledge, when facing the sample with a large uncertainty value, we adaptively increase the fusion weight of domain-specific knowledge ($\lambda_h$). Conversely, if the input has a low uncertainty value, the fusion weight of domain-agnostic knowledge ($\lambda_l$) will be increased. By employing the HKA strategy, our approach ensures that the adaptation process effectively captures relevant domain-specific knowledge while retaining long-term domain-shared knowledge.

### 3.4 Optimization Objective

Following previous CTTA work [57, 16], we leverage the teacher model $\mathcal{T}$ to generate the pseudo labels $\widetilde{y}$ for updating ViDAs. And we adopt consistency loss $L_{ce}$ as the optimization objective.

$$\mathcal{L}_{ce}(x) = -\frac{1}{C} \sum_{c}^{C} \widetilde{y}(c) \log \hat{y}(c) \tag{5}$$

Where $\hat{y}$ is the output of our student model $\mathcal{S}$, $C$ means the number of categories. Same as previous works[57, 16], we load the source pre-trained parameters to initialize the weight of both models and adopt the exponential moving average (EMA) to update the teacher model with ViDAs.

$$\mathcal{T}^t = \alpha \mathcal{T}^{t-1} + (1 - \alpha)\mathcal{S}^t \tag{6}$$

Where $t$ is the time step. And we set $\alpha = 0.999$ [52], which is the updating weight of EMA.

## 4 Experiment

In Section 4.2 and 4.3, we compare our method with other SOTA methods on classification and segmentation of CTTA. In Section 4.4, we employ the foundation model [32, 43] as the backbone and evaluate the efficacy of our method. In Section 4.5, we further evaluate the domain generalization ability of the proposed method. Comprehensive ablation studies are conducted in Section 4.6. More quantitative comparisons and qualitative analyses are shown in the supplementary materials.

### 4.1 Task settings and Datasets

**Dataset.** We evaluate our method on three classification CTTA benchmarks, including CIFAR10-to-CIFAR10C(standard), CIFAR100-to-CIFAR100C [33] and ImageNet-to-ImageNet-C [24]. For

segmentation CTTA [57, 59], we evaluate our method on Cityscapes-to-ACDC, where the Cityscapes dataset [11] serves as the source domain, and the ACDC dataset [50] represents the target domains.

**Baselines.** We compare the proposed method against two types of CTTA approaches, including (1)Modal-based: source model [14, 58], Pseudo-label [36], Tent-continual [56], CoTTA [57], and, SATA [7]. (2) Prompt-based: visual domain prompt [16].

**CTTA Task setting.** Following [57, 16], in classification CTTA tasks, we sequentially adapt the pre-trained source model to the fifteen target domains with the largest corruption severity (level 5). The online prediction results were evaluated immediately after encountering the input data. Regarding segmentation CTTA [57, 59], the source model [58] is an off-the-shelf pre-trained on the Cityscapes dataset [11]. As for the continual target domains, we utilize the ACDC dataset [50], which consists of images collected in four unseen visual conditions: Fog, Night, Rain, and Snow. To simulate continual environmental changes in real-life scenarios, we cyclically repeat the same sequence of target domains (Fog→Night→Rain→Snow) multiple times.

**Implementation Details.** In our CTTA experiments, we follow the implementation details specified in previous works [57, 59] to ensure consistency and comparability. we adopt ViT-base [14] and ResNet [22] as the backbone in classification CTTA. In the case of ViT-base, we resize the input images to 224x224, while maintaining the original image resolution for other backbones. For segmentation CTTA, we adopt the pre-trained Segformer-B5 model [58] as the source model. We down-sample the input size from 1920x1080 to 960x540 for target domain data [57]. The optimizer is performed using Adam [31] with $(\beta_1, \beta_2) = (0.9, 0.999)$. We set the learning rates to specific values for each backbone, such as 1e-5 for ViT and 3e-4 for Segformer. To initialize our visual domain adapters, we train the model with adapters for one epoch on the source domain. We apply a range of image resolution scale factors [0.5, 0.75, 1.0, 1.25, 1.5, 1.75, 2.0] for the augmentation method and construct the teacher model inputs [57]. All experiments are conducted on NVIDIA A100 GPUs.

## 4.2 The Effectiveness on Classification CTTA

Table 1: Classification error rate(%) for ImageNet-to-ImageNet-C online CTTA task. Gain(%) represents the percentage of improvement in model accuracy compared with the source method.

| Backbone | Method | REF | Gaussian | shot | impulse | defocus | glass | motion | zoom | snow | frost | fog | brightness | contrast | elastic_trans | pixelate | jpeg | Mean↓ | Gain |
|---|---|---|---|---|---|---|---|---|---|---|---|---|---|---|---|---|---|---|---|
| ResNet50 | Source [21] | CVPR2016 | 97.8 | 97.1 | 98.2 | 81.7 | 89.8 | 85.2 | 78 | 83.5 | 77.1 | 75.9 | 41.3 | 94.5 | 82.5 | 79.3 | 68.6 | 82 | 0.0 |
| | CoTTA [57] | CVPR2022 | 52.9 | 51.6 | 51.4 | 68.3 | 78.1 | 57.1 | 62.0 | 48.2 | 52.7 | 55.3 | 25.9 | 90.0 | 56.4 | 36.4 | 35.2 | 62.7 | +19.3 |
| | VDP [16] | AAAI2023 | - | - | - | - | - | - | - | - | - | - | - | - | - | - | - | 51.5 | +30.5 |
| | SATA [7] | 2023.4.20 | 74.1 | 72.9 | 71.6 | 75.7 | 74.1 | 64.2 | 55.5 | 55.6 | 62.9 | 46.6 | 36.1 | 69.9 | 50.6 | 44.3 | 48.5 | 60.1 | +21.9 |
| ViT-base | Source | ICLR2021 | 53.0 | 51.8 | 52.1 | 68.5 | 78.8 | 58.5 | 63.3 | 49.9 | 54.2 | 57.7 | 26.4 | 91.4 | 57.5 | 38.0 | 36.2 | 55.8 | 0.0 |
| | Pseudo [36] | ICML2013 | 45.2 | 40.4 | 41.6 | 51.3 | 53.9 | 45.6 | 47.7 | 40.4 | 45.7 | 93.8 | 98.5 | 99.9 | 99.9 | 98.9 | 99.6 | 61.2 | -5.4 |
| | Tent [56] | ICLR2021 | 52.2 | 48.9 | 49.2 | 65.8 | 73 | 54.5 | 58.4 | 44.0 | 47.7 | 50.3 | 23.9 | 72.8 | 55.7 | 34.4 | 33.9 | 51.0 | +4.8 |
| | CoTTA [57] | CVPR2022 | 52.9 | 51.6 | 51.4 | 68.3 | 78.1 | 57.1 | 62.0 | 48.2 | 52.7 | 55.3 | 25.9 | 90.0 | 56.4 | 36.4 | 35.2 | 54.8 | +3.6 |
| | VDP [16] | AAAI2023 | 52.7 | 51.6 | 50.1 | 58.1 | 70.2 | 56.1 | 58.1 | 42.1 | 46.1 | 45.8 | 23.6 | 70.4 | 54.9 | 34.5 | 36.1 | 50.0 | +5.8 |
| | **Ours** | **Proposed** | **47.7** | **42.5** | **42.9** | **52.2** | **56.9** | **45.5** | **48.9** | **38.9** | **42.7** | **40.7** | **24.3** | **52.8** | **49.1** | **33.5** | **33.1** | **43.4** | **+12.4** |
| Directly test after adaptation | | | | | | | | | | | | | | | | | → | Mean↓ | Gain |
| ViT-base | **Ours** | **Proposed** | **46.2** | **44.4** | **45.8** | **48.9** | **52.1** | **45.0** | **48.6** | **37.5** | **41.9** | **39.5** | **23.9** | **49.0** | **49.0** | **32.1** | **32.6** | **42.4** | **+13.4** |

**ImageNet-to-ImageNet-C.** Given the source model pre-trained on ImageNet, we conduct CTTA on ImageNet-C, which consists of fifteen corruption types that occur sequentially during the test time. Table .1 demonstrates that the majority of methods employing the ViT backbone achieve lower classification errors compared to those using the ResNet50 backbone. For ViT-base, the average classification error is up to 55.8% when we directly test the source model on target domains. In contrast, our method can outperform all previous methods, achieving a 12.4% and 6.6% improvement over the source model and previous SOTA method, respectively. Moreover, our method showcases remarkable performance across the majority of corruption types, highlighting its effective mitigation of error accumulation and its capability for continual adaptation. After completing the entire CTTA process, we evaluate the performance of our method on the seen target domains. As shown in Table 1, the performance degradation is observed in only 2 out of 15 corruption types. Additionally, we achieve an overall improvement of 1.0% in the average classification error. These findings demonstrate that our method successfully preserves continual domain-shared knowledge and avoids catastrophic forgetting during CTTA. In conclusion, our homeostatic ViDAs can extract the different domain knowledge and avoid CTTA main challenges simultaneously.

Table 2: Average error rate (%) for the standard CIFAR10-to-CIAFAR10C and CIFAR100-to-CIAFAR100C CTTA task. All results are evaluated on the ViT-base, which is fully pre-trained on the source domain dataset.

| Target | Method | Source | Tent | CoTTA | VDP | Ours |
|--------|--------|--------|------|-------|-----|------|
| Cifar10C | Mean↓ | 28.2 | 25.5 | 24.6 | 24.1 | **20.7** |
| | Gain↑ | 0.0 | +2.7 | +3.6 | +4.1 | **+7.5** |
| Cifar100C | Mean↓ | 35.4 | 33.2 | 34.8 | 35.0 | **27.3** |
| | Gain↑ | 0.0 | +2.2 | +0.7 | +0.4 | **+8.1** |

Table 3: Average error rate (%) for the CIFAR10-to-CIFAR10C CTTA task. All results are evaluated on the ViT-Base, which uses the pre-trained encoder parameter of foundation models (DINOv2 [43] and SAM [32]).

| Backbone | Method | Source | Tent | CoTTA | Ours |
|----------|--------|--------|------|-------|------|
| DINOv2 | Mean↓ | 25.0 | 21.7 | 29.3 | **20.2** |
| | Gain↑ | 0.0 | +3.2 | −4.3 | **+4.8** |
| SAM | Mean↓ | 39.3 | 37.5 | 39.4 | **34.1** |
| | Gain↑ | 0.0 | +1.8 | −0.1 | **+5.2** |

To further validate the effectiveness of our method, we conduct experiments on CIFAR10-to-CIFAR10C and CIFAR100-to-CIFAR100C. As illustrated in Table .2, in CIFAR10C, our approach achieved a 3.4% improvement compared to the previous SOTA model. We extend our evaluation to CIFAR100C, which comprises a larger number of categories in each domain. Our approach surpasses all previous methods, which show the same trend as the above CTTA experiments. Therefore, the results prove that our method mitigates the challenges posed by continual distribution shifts, regardless of the number of categories present in each domain.

## 4.3 The Effectiveness on Segmentation CTTA

Table 4: **Performance comparison for Cityscape-to-ACDC CTTA.** We sequentially repeat the same sequence of target domains three times. Mean is the average score of mIoU.

| Time | | | | | | $t \longrightarrow$ | | | | | | | | | | | | |
|------|-----|-----|-----|-----|-----|-----|-----|-----|-----|-----|-----|-----|-----|-----|-----|-----|-------|------|
| Round | | 1 | | | | | 2 | | | | | 3 | | | | | Mean↑ | Gain |
| Method | REF | Fog | Night | Rain | Snow | Mean↑ | Fog | Night | Rain | Snow | Mean↑ | Fog | Night | Rain | Snow | Mean↑ | | |
| Source [58] | NIPS2021 | 69.1 | 40.3 | 59.7 | 57.8 | 56.7 | 69.1 | 40.3 | 59.7 | 57.8 | 56.7 | 69.1 | 40.3 | 59.7 | 57.8 | 56.7 | 56.7 | / |
| TENT [55] | ICLR2021 | 69.0 | 40.2 | 60.1 | 57.3 | 56.7 | 68.3 | 39.0 | 60.1 | 56.3 | 55.9 | 67.5 | 37.8 | 59.6 | 55.0 | 55.0 | 55.7 | -1.0 |
| CoTTA [57] | CVPR2022 | 70.9 | 41.2 | 62.4 | 59.7 | 58.6 | 70.9 | 41.1 | 62.6 | 59.7 | 58.6 | 70.9 | 41.0 | 62.7 | 59.7 | 58.6 | 58.6 | +1.9 |
| DePT [20] | ICLR2023 | 71.0 | 40.8 | 58.2 | 56.8 | 56.5 | 68.2 | 40.0 | 55.4 | 53.7 | 54.3 | 66.4 | 38.0 | 47.3 | 47.2 | 49.7 | 53.4 | -3.3 |
| VDP [16] | AAAI2023 | 70.5 | 41.1 | 62.1 | 59.5 | 58.3 | 70.4 | 41.1 | 62.2 | 59.4 | 58.2 | 70.4 | 41.0 | 62.2 | 59.4 | 58.2 | 58.2 | +1.5 |
| **Ours** | **Proposed** | **71.6** | **43.2** | **66.0** | **63.4** | **61.1** | **73.2** | **44.5** | **67.0** | **63.9** | **62.2** | **73.2** | **44.6** | **67.2** | **64.2** | **62.3** | **61.9** | **+5.2** |

Figure 4: Qualitative comparison of our method with previous SOTA methods on the ACDC dataset. Our method could better segment different pixel-wise classes such as shown in the white box.

**Cityscapes-to-ACDC.** To demonstrate the effectiveness of our method in the semantic segmentation CTTA task, we conducted evaluations on four target domains from the ACDC dataset periodically during test time. As presented in Table 4, we observed a gradual decrease in the mIoUs of TENT and DePT over time, indicating the occurrence of catastrophic forgetting. In contrast, our method has a continual improvement of average mIoU (61.1→62.2→62.3) when the same sequence of target domains is repeated. Significantly, the proposed method surpasses the previous state-of-the-art CTTA method [57] by achieving a 3.3% increase in mIoU. This notable improvement showcases our method's ability to adapt continuously to different target domains in the pixel-level task. In Fig .4, our method correctly distinguish the sidewalk from the road, avoiding mis-classification.

## 4.4 Continual Adapting for Foundation Models

Foundation models [5] are trained on large-scale datasets, endowing them with powerful generalization capabilities and the ability to capture representations of common features. However, performing full fine-tuning on the foundation model is time-consuming and economically impractical. Hence, our adaptation method proves valuable by enhancing the continual transfer performance of foundation

Table 5: The domain generalization comparisons on ImageNet-C. Results are evaluated on ViT-base. Mean and Gain(%) represent the performance on unseen target domains.

| Method | Directly test on unseen domains | | | | | Unseen |
| | bri. | contrast | elastic | pixelate | jpeg | Mean↓ |
|---|---|---|---|---|---|---|
| Source | 26.4 | 91.4 | 57.5 | 38.0 | 36.2 | 49.9 |
| Tent | 25.8 | 91.9 | 57.0 | 37.2 | 35.7 | 49.5 |
| CoTTA | 25.3 | 88.1 | 55.7 | 36.4 | 34.6 | 48.0 |
| **Ours** | **24.6** | **68.2** | **49.8** | **34.7** | **34.1** | **42.3** |

Table 6: Average error rate (%) for the ImageNet-to-ImageNet-C. Results are evaluated on the ViT. $ViDA_h$ and $ViDA_l$ represent the ViDAs with high-rank and low-rank prototypes.

| | $ViDA_h$ | $ViDA_l$ | HKA | Mean↓ |
|---|---|---|---|---|
| $Ex_1$ | - | - | - | 55.8 |
| $Ex_2$ | ✓ | - | - | 51.2 |
| $Ex_3$ | - | ✓ | - | 50.7 |
| $Ex_4$ | ✓ | ✓ | - | 45.6 |
| $Ex_5$ | ✓ | ✓ | ✓ | 43.4 |

models. As indicated in Table. 3, we introduce foundation models as the pre-trained model and adapt them to continual target domains (CIFAR10C). Our approach achieved a performance improvement of 4.8% on the representative image-level foundation model DINOv2 [43] and 5.2% on pixel-level foundation model SAM [32]. Our method consistently and reliably improves the performance of the foundation model on the unseen continual target domains. Note that, we only use the pre-trained encoder of SAM and add a classification head, which is fine-tuned on the source domain. During the inference phase, the ViDAs with a linear relationship can be projected onto the pre-trained foundation model through re-parameterization. This process empowers the foundation model with the learned different domain representations and maintains the model plasticity.

## 4.5 Domain Generalization on Unseen Continual Domains

To investigate the domain generalization (DG) ability of our method, we follow the leave-one-domain-out rule [62, 38] to leverage 10/15 domains of ImageNet-C as source domains for model training while the rest (5/15 domains) are treated as target domains without any form of adaptation. Specifically, we first use our proposed method to continually adapt the pre-trained model to 10/15 domains of ImageNet-C without any supervision. Then we directly test on the 5/15 unseen domains. Surprisingly, our method reduces 7.6% on the average error on unseen domains (Table 5), which has a significant improvement over other methods. The promising results demonstrate that our method possesses DG ability by effectively extracting domain-agnostic knowledge. This finding provides a new perspective on enhancing DG performance. More DG experiments are provided in the supplementary materials.

## 4.6 Ablation study

**Effectiveness of each component.** We conduct the ablation study on ImageNet-to-ImageNet-C CTTA scenario and evaluate the contribution of each component in our method, including high-rank ViDA ($ViDA_h$), low-rank ViDA ($ViDA_l$), and Homeostatic Knowledge Allotment (HKA) strategy. As shown in Table .6, $Ex_1$ represents the performance of the source pre-trained model (only 55.8%). In $Ex_2$, by introducing the high-rank ViDA, the average error decrease 4.6%, demonstrating that the high-rank prototype can extract more domain-specific knowledge to adapt in target domains. As illustrated in $Ex_3$, low-rank ViDA gains 5.1% improvement compared to $Ex_1$. The result proves that the domain-share knowledge extracted from low-rank prototypes can also improve the classification ability on continual target domains. $Ex_4$ has a remarkable improvement of 10.2% overall, demonstrating that the two types of ViDA can compensate for each other in the continual adaptation process. $Ex_5$ achieves 12.4% improvement in total, showcasing the effectiveness of the HKA strategy in maximizing the CTTA potential of both types of ViDA.

## 5 Conclusion and Limitations

In this paper, we propose a homeostatic Visual Domain Adapter (ViDA) to address error accumulation and catastrophic forgetting problems in Continual Test-Time Adaptation (CTTA) tasks. And we investigate that the low-rank ViDA can disregard the impact of dynamic distribution shifts and prioritize the extraction of domain-invariant knowledge, and the high-rank ViDA can extract more reliable domain-specific knowledge. Meanwhile, we further propose a Homeostatic Knowledge Allotment (HKA) strategy to dynamically fuse the knowledge from each ViDA with different rank prototypes. For limitations, the injected ViDAs and teacher-student scheme brings extra parameters and computational costs during the continual adaptation process.

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
