# OpenReview forum: "ViDA: Homeostatic Visual Domain Adapter for Continual Test Time Adaptation"
_NeurIPS.cc/2023/Conference — Submitted to NeurIPS 2023_

### Official Review · Reviewer_56Wj · 2023-07-05

**Soundness:** 3 good
**Presentation:** 3 good
**Contribution:** 3 good
**Rating:** 6
**Confidence:** 4

**Summary:**

The paper addresses the problem of continual test time adaptation by proposing to add two branches of low-rank and high-rank adapters.
The paper claims that the low-rank adapter learns the domain-agnostic knowledge, whereas the high-rank adapter captures the domain-specific knowledge.
The paper also proposes a Homeostatic Knowledge Allotment (HKA) strategy to discern the contribution of the two branches.
Experiments are conducted for the classification problem on the benchmarks ImagenetC, CIFAR10C, and CIFAR100C, and for the semantic segmentation on the Cityscapes-to-ACDC benchmark.
The experimental results demonstrate the effectiveness of the proposed approach as it achieves state-of-the-art results using some selected pre-trained deep neural network architecture.

**Strengths:**

1. The paper tries to capture the domain-specific and domain-agnostic knowledge using two separate adapters: low-rank and high-rank adapters.
2. The HKA strategy to update the weight contribution of these adapters based on the uncertainty value of prediction is a nice idea.
3. Ablation study demonstrates the contribution of different components.
4. Experimental results for zero-shot generalization are new in the TTA setting.

**Weaknesses:**

1. No theoretical justification or even intuition is provided about why the low-rank adapter learns the domain-agnostic knowledge, whereas the high-rank adapter captures the domain-specific knowledge.
2. The backbone architecture is not consistent with previous works such as CoTTA. Section 3.4 in the supplementary material has only for CIFAR10C. What about the performance of the proposed approach for Imagenet-to-ImagenetC on ResNet50? What about the performance of the proposed approach for CIFAR100-to-CIFAR100C for ResNeXt-29 architecture?
3. Retraining the "model added with low/high-rank adapters" for some steps using the source data. This leads to the inability to use off-the-shelf pre-trained models without access to the source domain data.

Minor Comments
1. References should be conference/journal version, for example, 57 reference can cite the CVPR conference version
2. Line 175: Figure .2 --> Figure 2, and many more "Figure"
3. Line 237: Modal --> Model
4. Line 248: comparability. --> comparability

**Questions:**

1. Equation 4 depicts the formulation to modify the scale factors $\lambda_h$ and $\lambda_l$ based on uncertainty scores. What if the conditions to update the scale factors are reversed? In other words, when facing a sample with a high uncertainty value, decrease $\lambda_h$ (instead of increasing it), and vice versa for $\lambda_l$. It would be interesting to see what happens in such a scenario.
2. Why the low-rank adapter learns the domain-agnostic knowledge, whereas the high-rank adapter captures the domain-specific knowledge? Can the authors provide some intuition behind it, other than t-SNE plot of embeddings?
3. The backbone architecture for most experiments is not consistent with previous works such as CoTTA. Section 3.4 in the supplementary material has only for CIFAR10C. What about the performance of the proposed approach for Imagenet-to-ImagenetC on ResNet50 (missing in Table 1)? What about the performance of the proposed approach for CIFAR100-to-CIFAR100C for ResNeXt-29 architecture?
4. Why for the semantic segmentation task only the results for 3 rounds has been reported, as approaches such as CoTTA experiments for for 10 rounds?

**Limitations:**

1. The experimental results indicate the effectiveness of the proposed approach for transformer based architecture and not much for convolutional neural network based architecture.
2. Addition of extra modules, such as adapters, requires further training of the model with adapters for one epoch on the source domain data. This is a limitation since the source domain data may not be available in many practical scenarios, and only the pre-trained model may be provided.

---

> ### Author Rebuttal · Authors · 2023-08-09
>
> - Q1 'More intuitions of the low-rank adapter and high-rank adapter': Thank you for the constructive advice, please refer to the global rebuttal Q1, including the justifications of H-divergence verification[18], Class Activation Mapping (CAM) visualization, and long-term CTTA experiment.
> - Q2 'CNN backbone': Thank you for your comprehensive feedback. We will further extend our CTTA experiments using CNN backbones in the final version. The following Table 1 presents the results of our ImageNet-to-ImageNet-C CTTA experiment with ResNet50 as the backbone. Our approach substantially reduces the error rate by 20.8% compared with the source model and achieves competitive performance compared with other CTTA methods. Additionally, our CIFAR100-to-CIFAR100C CTTA experiment with the ResNeXt-29 backbone is detailed in the following Table 2. Our proposed method shows a promising result, further underscoring the effectiveness of our approach in addressing the CTTA challenge with a CNN backbone. Given the remarkable performance and generalization capability demonstrated by vision transformers in visual recognition tasks [14], we leverage transformer models as the backbone on classification and segmentation CTTA settings. Meanwhile, since parameter-efficient fine-tuning (PEFT) methods (i.e., adapter, prompt) are more adaptive for transformer architecture [10,20,29], we exhibit transformer-based CTTA results in the main experiment.
>
> | Method(ResNet50) |   Source | BN adapt[31] | Tent[35] | CoTTA[36] | Ours |
> | --- | --- | --- | --- | --- | --- |
> | Average error rate | 82.0 | 68.6 | 62.6 | 62.7 | 61.2(+20.8%) |
>
>
>
> | Method(ResNeXt-29) |   Source | BN adapt[31] | Tent[35] | CoTTA[36] | Ours |
> | --- | --- | --- | --- | --- | --- |
> | Average error rate |   46.4 | 35.4 | 60.9 | 32.5 | 31.5(+14.9%) |
>
>
> - Q3 'Pre-train on source data': In Lines 254-255, we pre-train the initial adapter in the source domain for several iterations before adapting to continual target domains, aiming to establish a stable initial parameter base for ViDAs. However, this pre-training step is optional and doesn't compromise the effectiveness of our approach. Illustrated in the following Table (Cityscape-to-ACDC CTTA), ViDAs exhibit notable enhancements on the CTTA challenge even with random initial parameters or parameters pre-trained on ImageNet. To better showcase the practicality, we will add the experiments of other parameter initialization techniques in the final version.
> |  | Adapter pretrain | Fog | Night | Rain | Snow | Mean (IoU) |
> | --- | --- | --- | --- | --- | --- | --- |
> | Source [58] | - | 69.1 | 40.3 | 59.7 | 57.8 | 56.7 |
> | CoTTA [57] | - | 70.9 | 41.2 | 62.4 | 59.7 | 58.6 |
> | Ours | Source | **71.6** | 43.2 | **66.0** | 63.4 | 61.1 |
> | Ours | Random initial | **71.6** | 43.6 | 64.9 | 61.9 | 60.5 |
> | Ours | ImageNet | **71.6** | **44.3** | **66.0** | **63.5** | **61.4** |
>
> - Q4 'Inversed HKA': In response to the review comments, we conduct an additional experiment to invert the scale factors within the Homeostatic Knowledge Allotment (HKA) strategy. Specifically, for samples exhibiting high uncertainty, we reduce λh while increasing λl. As shown in the following Table, building upon the ablation study (Table 6) of the main paper, we integrate the **Inversed HKA** approach into Ex4, which already incorporates both low-rank and high-rank ViDAs. This adaptation yields an average error rate of 46.3 on ImageNet-C, marking a slight 0.7% error rate increase compared to Ex4. This experiment underscores how the proposed HKA strategy promotes the different domain representations between low-rank and high-rank VIDA models.
> |   | Contributions |   Average error rate |
> | --- | --- | --- |
> | Ex4 | ViDAh + ViDAl | 45.6 |
> | Ex5 | ViDAh + ViDAl + HKA | 43.4 |
> | **Inversed HKA** | ViDAh + ViDAl + Inversed HKA | **46.3** |
>
> - Q5 '10 rounds CTTA': We further present the segmentation CTTA experiment with 10 rounds on the rebuttal PDF Table 2. Notably, it demonstrates a consistent enhancement in mean mIoU during the initial rounds (rounds 1-3) while maintaining stable performance in subsequent rounds. Meanwhile, we also provide 10 rounds of classification CTTA experiment on rebuttal PDF Figure 2. Our proposed method (red line) shows a consistently declined average error rate during the long-term adaptation process, demonstrating its effectiveness and stability in addressing continual domain shifts. Finally, we will address minor comments and incorporate the comprehensive 10 rounds experiment into the final version.

---

> > ### Comment · Reviewer_56Wj · 2023-08-15
> > **Most of the Questions are Addressed in the Rebuttal**
> >
> > I want to thank the authors for addressing my comments and questions.
> > Other reviewers had similar questions regarding the utility of low-rank/high-rank adapters, which the authors have addressed to some extent in the rebuttal.
> >
> > The additional experimental results in the rebuttal shows the proposed approach's effectiveness.
> > Inversed HKA results empirically support the intuition that the high-rank adapter focuses on domain-specific knowledge, and the low-rank adapter focuses on domain-agnostic knowledge.
> >
> > However, the visualization of CAM, similar to Figure 3, for Inversed HKA would have further supported the claim.
> >
> > Based on the response, I do not have any further queries.
> >
> > Thank you.

---

> > > ### Author Response · Authors · 2023-08-15
> > > **Response to Reviewer 56Wj**
> > >
> > > Dear Reviewer 56Wj,
> > >
> > > We greatly appreciate your valuable feedback and the acknowledgment of our intuition that the high-rank adapter emphasizes domain-specific knowledge, while the low-rank adapter focus on domain-agnostic knowledge. Your crucial suggestion to introduce an inverted HKA experiment is immensely appreciated, and we are committed to integrating these additional rebuttal experiments into the final version.
> > >
> > > Unfortunately, due to restrictions at the current stage, we are unable to include figures or anonymous links in our official comments. However, rest assured that in the final version, we promise to add the CAM result of the inverted HKA to further substantiate our contributions. Thank you so much for your time and insightful comments.
> > >
> > > Best Regards

---

### Official Review · Reviewer_dPMr · 2023-07-05

**Soundness:** 3 good
**Presentation:** 3 good
**Contribution:** 3 good
**Rating:** 5
**Confidence:** 4

**Summary:**

This paper proposes to utilize domain-specific and domain-agnostic knowledge to tackle the error accumulation and catastrophic forgetting problem and boost the performance of continually test-time adaptation task. The proposed visual domain adaptor (ViDA) aims to adapt current domain distribution and maintain the continual domain-shared knowledge in CTTA, while a homeostatic knowledge allotment strategy is designed to fuse the knowledge. The experiments demonstrate that the method achieved SOTA.

**Strengths:**

1. The logic of this paper is clear and the performance is godd.
2. The whole method is easy to understand and implement.
3. They conduct sufficient experiments to prove their proposal.


**Weaknesses:**

1. The authors mentioned that the proposed method ensures no extra parameter increase in Section 1 (L64) and Section 3 (L195). However, extra parameters and computational costs are considered as limitations in L335, which brings contradiction and unclarity. Adding clear explanations would be appreciated.
2. The authors argues that ViDA with a low-rank prototype focuses on domain-agnostic knowledge while ViDA with a high-rank prototype concentrates more on domain-specific knowledge. Explanation on the Fig.1 is not sound enough.
3. How are the different ViDAs projected into the pre-trained model by re-parametrizations? Please provide more details.
4. Other suggestions：
 L261 Table .1 -> Table 1
 L275 Table .2 -> Table 2
 L296 Table. 3 -> Table 3


**Questions:**

Please refer to the weaknesses.

**Limitations:**

Limitations on extra computational costs are discussed.

---

> ### Author Rebuttal · Authors · 2023-08-09
>
> - Q1 'Extra parameter': We appreciate your insightful feedback. In the final version, we will provide clearer explanations of parameter usage. Regarding Lines 64 and 195, we elaborate on the fact that ViDAs can be re-parameterized and projected into the original model due to their linear relationship, ensuring no additional parameters for the **network**. Different from the network parameters discussion, in Line 335, we aim to express that our entire **training framework** (teacher-student model) will increase the extra computational costs during test-time parameters optimization.
> - Q2 'More explanation of different domain representation for ViDAs': Thank you for the constructive advice, please refer to the global rebuttal Q1, including the justifications of the H-divergence verification[18], Class Activation Mapping (CAM) visualization, and long-term CTTA experiment.  Besides, we design a verification experiment using adapters with the same structures, aiming to demonstrate the necessity of low-rank ViDA and high-rank ViDA. We have executed an ImageNet-to-ImageNet-C CTTA experiment using a combination of two high-rank adapters or two low-rank adapters, as shown in the rebuttal PDF Table 1. To ensure fairness, we conducted these experiments without implementing the homeostatic knowledge allotment (HKA) strategy. Notably, the two low-rank adapters (Ex2) demonstrated consistently lower long-term error rates compared to the source model and two high-rank adapters. Above results can be attributed to the fact that the two low-rank ViDAs tend to learn more general information and domain-invariant knowledge during continual adaptation. However, our method outperforms the two low-rank adapters across 14 out of 15 corruption types. This indicates that solely relying on low-rank adapters without the involvement of high-rank adapters is insufficient to fit target domains and match their data distribution. On the other hand, the performance of the two high-rank adapters initially surpasses ours (Ex4) in the early stages, covering the first few target domains. Nevertheless, a noticeable performance degradation becomes apparent in later target domains. This observation underscores a crucial finding: while increasing the number of high-rank ViDAs might enhance domain-specific knowledge acquisition during the initial phases of CTTA, it simultaneously exacerbates catastrophic forgetting throughout the entire adaptation process. In contrast, the fusion of both low-rank and high-rank ViDAs (Ex4) yields the most substantial improvement when compared to other configurations. This collaborative approach leverages the distinct domain representations of these adapters to compensate for each other's advantages and achieve a more robust and effective continual adaptation strategy.
> - Q3 'Re-parametrizations': Following the linear design of the adapter, we integrate trainable ViDAs into the original model, ensuring no additional parameters burden the network. Since fully connected layers of  low-rank and high-rank ViDAs are of linear relationship, the re-parametrization process for both is identical. Specifically, we denote the parameter matrices Wup and Wdown to represent the up-projection and down-projection linear layers, respectively.  The adapter's output Ya is computed as Ya = Wup(Wdown(x)), where x is the input. Given the linear relationship between Wup and Wdown, we construct a composite parameter matrix Wa, where Wa = Wup x Wdown. Consequently, the adapter's output can be expressed as Ya = Wa(x). Meanwhile, Wo signifies the parameter matrix of the original model's fully connected layer. The fused output Yf is computed as Yf = Wo(x) + Wa(x), or succinctly as Yf = (Wo + Wa)(x). Analogously, due to the linear relationship between Wo and Wa , the new parameter matrix Wf (Wf = Wo + Wa) is constructed to re-express the fused output Yf = Wf(x). Finally, we will fix all the suggestions in the final version.

---

> ### Author Response · Authors · 2023-08-17
> **Looking Forward to Seeing Your Response!**
>
> Dear reviewer dPMr,
>
> As the discussion phase is quickly passing, we want to know if you have any further questions or suggestions, and we are more than happy to discuss. Thanks again for your valuable reviews!
>
> Best, All anonymous authors

---

> ### Author Response · Authors · 2023-08-21
> **Waiting For Your Response**
>
> Dear Reviewer dPMr,
>
> With the discussion phase swiftly progressing, we seek to confirm if our response addresses your concerns. Feel free to inquire about any remaining questions, and we'll provide prompt responses.  If your concerns have been addressed, we would greatly appreciate it if you considered raising your score. Thank you for your valuable time and insightful comments.
>
> Best, All anonymous authors

---

### Official Review · Reviewer_DEpm · 2023-07-07

**Soundness:** 2 fair
**Presentation:** 2 fair
**Contribution:** 2 fair
**Rating:** 6
**Confidence:** 4

**Summary:**

This paper aims to address continual test-time adaptation (CTTA) with parameter-efficient fine-tuning techniques, i.e., adapter. The authors find that the low-rank adapter i.e., standard bottleneck structure, can extract domain-invariant knowledge. On the other hand, a high-rank adapter can extract more domain-specific knowledge. Experiments on three image classification benchmarks and one semantic segmentaiton benchmark demonstrate the effectiveness of the proposed method based on ViT backbone.

**Strengths:**

1. The paper is easy-to-follow.

2. The topic is essential yet the idea is moderate. Both continual test-time adaptation and parameter-efficient fine-tuning are important topics for the community and the paper addresses two formulations simultaneously.

3. A new method that seems well-motivated and performs well on classification and segmentation benchmarks.

**Weaknesses:**

1. The major concern is that are the learned features domain-invariant related to the structure of the adapter? That is, low-rank adapter can learn domain-invariant features while high-rank adapter can learn domain-specific features. How about all adapters adopt the same structures?

2. What is the motivation of the teacher model? Just following CoTTA?

3. More relevant works [a,b,c, d, e] should be discussed.

4. What is the design of low-rank and high-rank adapter in CNN architecture? Why not the auhors report the result of ResNet50 in Table 1?

5. It seems that the proposed ViDA module is memory-extensive since all experiments are conducted on A100 GPU. However, the backbone is just ResNet50 or ViT-base. Thus, does it mean that PEFT is memory-extensive or the proposed ViDA module is extensive?

6. It can be seen from the results in Table 3 that the model performance with the pre-trained encoder parameters of SAM significantly decreases. But the reason is unclear. It might be that SAM is pixel-level foundation model but the performance gains with image-level foundation model DINOv2 are limited. Also, I am interested in the results for Cityscapes-to-ACDC with SAM pre-trained parameters.

Refs:
[a] Niu et al., Towards stable test-time adaptation in dynamic wild world.
[b] Yuan et al., Robust test-time adaptation in dynamic scenarios.
[c] Song et al., EcoTTA: Memory-Efficient Continual Test-time Adaptation via Self-distilled Regularization.
[d] Gong et al., NOTE: Robust continual test-time adaptation against temporal correlation.
[e] Döbler et al., Robust mean teacher for continual and gradual test-time adaptation.


**Questions:**

1. What do t-SNE results of other blocks look like? Why the authors select the third transformer block to analye in Fig. 1 (b).

2. What is the meaning of ``prototype``?

3. Minor:

-  ref [55] and ref [56] are duplicated.

**Limitations:**

The authors claimed one limitation of this work, i.e., introducing extra parameters and computational cost.

---

> ### Author Rebuttal · Authors · 2023-08-09
>
> - Q1 'Different domain representation of ViDAs': Thank you for the comprehensive comments. The experiment analysis of all adapters with the same structure is shown in Reviewer#vo5W Q3 while more justifications are illustrated in the global rebuttal Q1.
> - Q2 'Why teacher model': Motivated by the fact that the mean teacher predictions have a higher quality than the standard model [52], we utilize a mean-teacher model to provide more accurate predictions in the continual adaptation process. Meanwhile, since [e] observes that mean teachers are more robust in dynamic environments, we leverage the teacher model to maintain the stability and previous domain knowledge during continual test-time adaptation (CTTA). Besides, we aim to conduct a fair comparison with previous CTTA works [16,57], thus we utilize a similar teacher-student framework in our pipeline. Notably, the flexibility of ViDA allows integration with various optimization methodologies, like Tent [56].
> - Q3 'More related discussion': References [a] and [b] focus on tackling TTA in dynamic scenarios, which is different from traditional CTTA addressed by [16,57,c]. Specifically, [a] demonstrates the benefits of batch-agnostic norm layers compared to BN under wild test settings and addresses model collapse, however, it doesn't directly tackle the continual shift challenge. [b] introduces a robust BatchNorm layer for Practical Test-Time Adaptation (incorporating distribution changes and correlation sampling), but ignores catastrophic forgetting in dynamic environments. Notably, [d] introduces instance-aware normalization and prediction-balanced reservoir sampling for stable adaptation but it does not explicitly consider instability facing long-term shifting distributions. In contrast, Ecotta [c] introduces a meta-network for the CTTA problem, aiming to avoid error accumulation by regularizing the outputs from meta-network and frozen network. However, a significant portion of the frozen network only retains source domain knowledge, overlooking target domain knowledge. On the other hand, RMT [e] tackles error accumulation via gradient analysis and introduces a symmetric cross-entropy loss for CTTA. Our approach diverges from these by employing a novel adapter-based adaptation scheme that adopts different domain representations of low-rank and high-rank ViDA to explicitly tackle the catastrophic forgetting and error accumulation problems. As shown in the following Table, our method achieves superior performance compared to RoTTA[b], NOTE [d], and EcoTTA [c] on Cifar10-C with WideResNet-28 backbone. Due to ViDA is a parameter-efficient and plug-and-play method, it has the potential for integration with other methods to collectively tackle the challenge of continual distribution shift.
> | Cifar10-C(WideResNet-28) | RoTTA[b] | NOTE [d] | EcoTTA [c] | RMT [e] | Ours |
> | --- | --- | --- | --- | --- | --- |
> | Average error rate | 17.5 | 20.2 | 16.8 | 14.5 | 15.8 |
>
>
> - Q4 'CNN architecture': In Lines 181-183 of the main paper and Lines 111-114 of the supplement, instead of employing linear layers, we have employed 1 × 1 convolutional layers to construct both down-projection and up-projection layers for CNN backbones. As shown in the following table, we carry out an ImageNet-to-ImageNet-C CTTA experiment with a ResNet50 backbone.  Our method achieves a 61.2% average error rate, obtaining a substantial decrease of 20.8% compared to the source model. Meanwhile, the CIFAR100-to-CIFAR100C CTTA experiment with the ResNeXt-29 backbone also shows the effectiveness of our approach. These findings show the robust performance of our method in addressing the challenges of the CTTA problem with a CNN backbone. Given the remarkable performance and generalization capability demonstrated by vision transformers [14], we select our CTTA settings to employ transformer backbones.
> | Method(ResNet50) |   Source | BN adapt[31] | Tent[35] | CoTTA[36] | Ours |
> | --- | --- | --- | --- | --- | --- |
> | Average error rate | 82.0 | 68.6 | 62.6 | 63.0 | 61.2(+20.8%) |
>
> | Method(ResNeXt-29) |   Source | BN adapt[31] | Tent[35] | CoTTA[36] | Ours |
> | --- | --- | --- | --- | --- | --- |
> | Average error rate |   46.4 | 35.4 | 60.9 | 32.5 | 31.5(+14.9%) |
>
>
> - Q5 ' Extensive': When utilizing backbone such as ResNet50 or ViT-base, ViDA is not memory-extensive. In our ImageNet-to-ImageNet-C CTTA experiment, our pipeline utilizes only 18GB out of the available 80GB GPU memory on A100 (image resolution = 224x224 and  batch size = 32).
> - Q6 'SAM': We conduct segmentation CTTA with SAM pre-trained parameters on the Cityscapes-to-ACDC scenario. However, it's worth noting that Segformer [58], which we used in the main experiments, does not incorporate positional encoding. We thus adopt the SETR [f] model as our new baseline to load SAM's pre-trained parameters. As shown in the following Table, Our approach with SAM pre-trained parameters outperforms others on the ACDC target domains. This aligns with your assumption: SAM, being a pixel-level foundational model, excels in capturing fine-grained feature representations in dense CTTA tasks. [f] Rethinking semantic segmentation from a sequence-to-sequence perspective with transformers
> |  | Pretrained | Fog | Night | Rain | Snow | Mean (IoU) |
> | --- | --- | --- | --- | --- | --- | --- |
> | Source | Source model | 72.6 | 43.1 | 63.0 | 64.3 | 60.8 |
> | Source | SAM | 74.8 | 44.1 | 66.7 | 66.6 | 63.0 |
> | Cotta | SAM | 75.4 | 45.9 | 67.3 | 68.7 | 64.3 |
> | Ours | SAM | **76.5** | **47.2** | **68.1** | **70.7** | **65.6** |
>
> - Q7 't-SNE': Due to space limitation, we analyze the middle layer of the backbone in our t-SNE study. Additional t-SNE results for the first and last transformer blocks are available in rebuttal PDF Figure 1, showing the same tendency as the results from the third transformer block.
> - Q8 'Prototype': The prototype refers to the intermediate feature latent space of ViDAs. And we will address the 'Minor' issue in the final version.

---

> > ### Comment · Reviewer_DEpm · 2023-08-12
> > **Further discussion**
> >
> > Dear authors,
> >
> > I really appreciate the authors' responses. I have read your rebuttal and  almost of my concerns have been solvedand. Here are my comments.
> >
> > First, I am indeed surprised at the ability of different structures (low-rank & high-rank) to learn different features (domain-invariant &domain-specific). The authors should have emphasised these interesting findings in the manuscript.
> >
> > Second, with the rapid development of the field of test-time adaptation in recent two years, the authors should pay more attention to the latest works and then discuss and compare them as much as possible to highlight the strengths of this paper.
> >
> > Thirdly, I am very grateful for the results provided in the rebuttal phase and hope to see more analyses in the camera-ready version! For example, integration with other methods and experiments on ResNet, etc.
> >
> > Certainly, my current inclination is to extend my support to this paper, and have revised my rating.

---

> > > ### Author Response · Authors · 2023-08-13
> > > **Further Discussions for Reviewer DEpm**
> > >
> > > Dear reviewer DEpm，
> > >
> > > We appreciate your valuable feedback and the acknowledgment of our proposed method.
> > >
> > > Firstly, we will elaborate on the justifications for the diverse domain representations of low-rank ViDA and high-rank ViDA in the manuscript, as evidenced in our rebuttal responses.
> > >
> > > Secondly, we would like to extend our gratitude for your insightful suggestions for improvement. We intend to delve deeper into the latest test-time adaptation approaches (e.g., [a][b][c][d][e]) and establish a comprehensive comparison to underscore our method's strengths in the final version.
> > >
> > > Furthermore, we are committed to incorporating the rebuttal experiments and providing in-depth analysis in the camera-ready version. It's important to note that we will particularly focus on refining the CNN-based CTTA experiments and validating the seamless integration capability of our proposed method.
> > >
> > > Tank you again for your time and insightful comments.

---

### Official Review · Reviewer_vo5W · 2023-07-08

**Soundness:** 2 fair
**Presentation:** 3 good
**Contribution:** 2 fair
**Rating:** 5
**Confidence:** 5

**Summary:**

This paper proposes a continual test-time adaptation method by designing a visual domain adapter (ViDA) to handle both domain-specific and domain-agnostic knowledge. To adapt to different distribution shifts, a homeostatic knowledge allotment strategy is proposed to adaptively merge knowledge from each ViDA with different rank prototypes. Experiments on four benchmark datasets demonstrate the effectiveness of the proposed method for both classification and segmentation CTTA tasks.

**Strengths:**

- The paper introduce homeostatic visual domain adapter for continual test time adaptation.
- Good results are achieved on both classification and segmentation tasks.

**Weaknesses:**

- In general, the proposed components are not very well justified and verified. The details are listed in the following points.
- Except for the empirical results, it lacks more in-depth analyses and justifications on why a low-rank prototype focuses on domain-agnostic features while a high-rank prototype focuses on domain-specific knowledge. In addition, Figure 3 (a) shows the reduction of inter-domain divergence due to the introduction of the low-rank adapters. The reviewer was wondering what would be the results of the inter-domain divergence by using the high-rank adapters.
- In table 1, the experimental results are not clearly explained. For example, what are the difference between the last two rows?
- In Table 6, both the ViDAh and ViDAl could improve the performance significantly. However, it is still unclear why the different adapters (high-rank and low-rank) necessary. How to effectively show that they are complement to each other? It would be interesting to know whether two high-rank adapters or two low-rank adapters could also achieve good results?

**Questions:**

- Why a low-rank prototype focuses on domain-agnostic features while a high-rank prototype focuses on domain-specific knowledge?
- How to effectively show that ViDAh and ViDAl are complement to each other?

**Limitations:**

The authors addressed some of the limitations.

---

> ### Author Rebuttal · Authors · 2023-08-09
>
> - Q1: 'Different domain representation of low-rank ViDA and high-rank ViDA': Thank you for the constructive advice, please refer to the global rebuttal Q1, including the justifications of H-divergence verification[18], Class Activation Mapping (CAM) visualization, and long-term CTTA experiment.
> - Q2 ' More explanations of Table 1': Thank you for the detailed comments, we will add more explanations of Table 1 in the camera-ready version. The second row from the bottom of the table represents the standard ImageNet-to-ImageNet-C Continual Test-Time Adaptation (CTTA) experiment using our proposed method. Evidently, our approach outperforms both the source model and the prior state-of-the-art (SOTA) method, showing the effectiveness of our approach in addressing continual domain shifts. In reference to the last row, please refer to Lines 150-152 and Lines 267-268 for the explicit description of the experimental setup. This setup has been devised to validate our method's capability to mitigate the problem of catastrophic forgetting. Specifically, we preserve the final parameters of both the model and ViDAs after completing a round of CTTA. Subsequently, we utilize these fixed parameters to evaluate the model's performance across all previously encountered target domains within ImageNet-C, without further parameter updates. As illuminated in Table 1, Our method showcases an improvement of 1.0% in the average classification error compared to the result presented in the second row from the bottom. This outcome robustly substantiates that our approach effectively avoids the problem of catastrophic forgetting within the context of a continually changing environment.
> - Q3 ‘Complement each other’: Thank you for your valuable insights. We have executed an ImageNet-to-ImageNet-C CTTA experiment using a combination of two high-rank adapters or two low-rank adapters, as shown in the rebuttal PDF Table 1. To ensure fairness, we conducted these experiments without implementing the homeostatic knowledge allotment (HKA) strategy. Notably, the two low-rank adapters (Ex2) demonstrated consistently lower long-term error rates compared to the source model and two high-rank adapters. Above results can be attributed to the fact that the two low-rank ViDAs tend to learn general information and domain-invariant knowledge during continual adaptation. However, our method outperforms the two low-rank adapters across 14 out of 15 corruption types. This indicates that solely relying on low-rank adapters without the involvement of high-rank adapters is insufficient to fit target domains and match their data distribution. On the other hand, the performance of the two high-rank adapters initially surpasses our approach (Ex4) in the early stages, covering the first few target domains. Nevertheless, a noticeable performance degradation becomes apparent in later target domains. This observation underscores a crucial finding: while increasing the number of high-rank ViDAs might enhance domain-specific knowledge acquisition during the initial phases of CTTA, it simultaneously exacerbates catastrophic forgetting throughout the entire adaptation process. In contrast, the fusion of both low-rank and high-rank ViDAs (Ex4) yields the most substantial improvement when compared to other configurations. Our collaborative approach leverages the distinct domain representations of these adapters to compensate for each other's advantages and achieve a more robust and effective continual adaptation strategy.

---

> > ### Comment · Reviewer_vo5W · 2023-08-17
> > **Post rebuttal**
> >
> > The authors have conducted additional extensive experiments and most of my concerns are addressed. So I would like to update my rating and lean towards acceptance.

---

> > > ### Author Response · Authors · 2023-08-17
> > > **Response to Reviewer vo5W**
> > >
> > > Dear Reviewer vo5W,
> > >
> > > We greatly appreciate your valuable feedback and the acknowledgment of our proposed method that introducing high-rank and low-rank adapters to extract domain-specific and domain-invariant knowledge in the CTTA problem, respectively. And we are committed to incorporating the additional experiments and providing in-depth analysis in the camera-ready version.
> > >
> > > Best, All anonymous authors

---

> ### Author Response · Authors · 2023-08-17
> **Looking Forward to Seeing Your Response!**
>
> Dear Reviewer vo5W,
>
> Given the discussion phase is quickly passing, we want to know if our response resolves your concerns. If you have any further questions, we are more than happy to discuss them. Thanks again for your valuable suggestions!
>
> Best, All anonymous authors

---

### Author Rebuttal · Authors · 2023-08-09

**To ALL:**
 - Q1. 'Different domain representations of low-rank ViDA and high-rank ViDA'.
   -  Thank you for the comprehensive comments, and we will add the in-depth analyses and justifications in the final version, including different domain representations of low-rank and high-rank ViDAs. In section 3.1 and Figure 3 (a) of the main paper, according to previous domain transfer research (DANN [18]), we employ the **H-divergence metric** to evaluate the domain representations of ViDAs across different target domains. Following the suggestion of Reviewer vo5W, “verification the inter-domain divergence by using the high-rank adapter”, we supplement the results in the rebuttal PDF Figure 4, where the visual representation showcases the different knowledge extraction of ViDAs. These experiments have been conducted within the CIFAR10-to-CIFAR10C Continual Test-Time Adaptation (CTTA) scenario. As shown in the rebuttal PDF Figure 4, the feature representation produced by the high-rank adapter displays a more pronounced divergence when compared to the output of the low-rank adapter. It is important to note that when dealing with later target domains or encountering substantial domain shifts between two adjacent domains (as seen in the case of target domains 9-13), the inter-domain divergence of the high-rank adapter shows a notably higher value than others and presents a close divergence value compared with the original source model. In conjunction with the lower intra-class divergence presented in Figure 3(b) of submission, it is evident that while the high-rank adapter effectively facilitates domain-specific knowledge extraction in the current target domain, it simultaneously forfeits some of the domain knowledge acquired from preceding domains. In contrast, as shown in the rebuttal PDF Figure 4, the feature representation generated by the low-rank adapter continually demonstrates lower divergence compared to the others. Since the low-rank structure minimizes the redundancy within the feature representation, it compels the model to concentrate more on task-relevant information and mitigate the problem of catastrophic forgetting within the continual adaptation process.
   - In addition, we have extended our analysis by incorporating the visualization of **Class Activation Mapping (CAM)** on the ImageNet-to-ImageNet-C CTTA scenario. Specifically, we adopt CAM to compare the attention of the low-rank branch, high-rank branch, and the original model during the continual adaptation process. To elaborate, as shown in the rebuttal PDF Figure 3, we showcase the feature representation of the images from different target domains, including the noise of Gaussian and defocus blur. Our observations reveal that the low-rank ViDA is inclined to put more weight on the foreground sample while tending to disregard background noise shifts. This indicates that the low-rank ViDA attends to locations with more general and domain-agnostic information from target domains. Conversely, the high-rank ViDA exhibits an inverse pattern, as illustrated in the last column of rebuttal PDF Figure 3. It allocates more attention to locations characterized by substantial domain shift, encompassing the entirety of the input images. This behavior aligns with the high-rank branch's tendency to fit global information and predominantly extract domain-specific knowledge from the target domain data.
   - In order to provide further substantiation for the distinct domain representations of the low-rank and high-rank ViDAs, we have executed a **10 rounds CTTA experiment** on ImageNet-to-ImageNet-C. The outcomes of this experiment can be accessed through the rebuttal PDF Figure 2. In this comprehensive experiment, we simulate a long-term adaptation scenario by repeating 10 rounds of 15 corruption sequences in the ImageNet-C. Remarkably, the high-rank ViDA achieves competitive results over other methods during the initial 1 to 3 rounds. This result demonstrates the high-rank feature's capacity to efficiently learn target domain-specific knowledge. However, an increment in error rates becomes obvious during the later rounds (rounds 5 to 10). The results validate the potential for encountering catastrophic forgetting when focusing exclusively on domain-specific knowledge. In contrast, the performance of the low-rank ViDA remains consistently robust throughout the continual adaptation process, verifying it concentrates more on extracting task-relevant knowledge and effectively prevents the catastrophic forgetting problem.
   - In conclusion, our analysis and justifications support our hypothesis concerning the different domain representations of low-rank and high-rank ViDAs. In the case of low-rank ViDA, its structure reduces feature redundancy, leading to an underfit state during CTTA. As a result, it leans towards acquiring general information across continuous target domains, extracting domain-agnostic knowledge to mitigate catastrophic forgetting. In contrast, high-rank ViDA employs a higher-dimensional feature representation that better matches the data distribution in target domains, focusing on learning domain-specific knowledge to prevent error accumulation. Importantly, the specialized structures of low-rank and high-rank ViDAs contribute to distinct domain representation learning, which we further control using the Homeostatic Knowledge Allotment (HKA) strategy. For instance, when meeting the sample with a large distribution shift, we will increase the fusion weight of the high-rank feature (Equation 4), aiming to extract more domain-specific knowledge through the high-rank ViDA.

---

### Comment · Area_Chair_1U7o · 2023-08-13
**Discussion stage starts**

Dear Reviewers,

Thank you for reviewing this paper. Authors have provided their rebuttal. Would you please check it, and give your comments/rating based on the rebuttal letter and the comments from other reviewers.

Best Regards
AC

---

### Decision · Program_Chairs · 2023-09-21

**Decision:**

Reject

**Comment:**

This work introduces a homeostatic visual domain adapter for continual test time adaptation, and as mentioned by reviewers, both continual test-time adaptation and parameter-efficient fine-tuning are important topics, while this work addresses both formulations. This work leverages teacher-student learning paradigm, and aims to simultaneously exploit domain-specific and domain-agnostic knowledge for effective test time adaptation. Yet, the concept of leveraging domain specific and domain invariant information has been explored in the domain adaptation and domain generalization areas. A Homeostatic Knowledge Allotment scheme is designed employing the variance information for handling continual adaptation. Though the whole framework seems to be reasonable, the novelty of this work is not very obvious. Reviewers also pointed out that the work lacks theoretical analysis, and the design motivations of some modules (e.g., the teacher model) need to be further elaborated.  Thus two of the reviewers' comments tend to be positive while the remaining two gave borderline scores. AC and SAC have discussed extensively over this paper, and decide to reject this paper, considering that in terms of machine learning technical contributions, this work mainly rely on existing techniques and concepts, without brining significant novelties. AC and SAC also agree that there are merits in the problems solved and the design of the method. However, due to the weaknesses of this paper in technical novelties, motivation descriptions, and theoretical analysis, and also the high standard of this conference, the AC suggests the authors to further improve the work, such that the work can be resubmitted to a conference at the same level.